# Depolymerization of robust polyetheretherketone to regenerate monomer units using sulfur reagents

Yasunori Minami [1,2✉], Nao Matsuyama[1], Yasuo Takeichi [3], Ryota Watanabe [4], Siby Mathew[1] & Yumiko Nakajima [1]

Super engineering plastics, high-performance thermoplastic resins such as polyetheretherketone, and polyphenylene sulfide have been utilized in industries, owing to their high thermal stability and mechanical strength. However, their robustness hinders their depolymerization to produce monomers and low-weight molecules. Presently, chemical recycling for most super engineering plastics remains relatively unexplored. Herein, we report the depolymerization of insoluble polyetheretherketone using sulfur nucleophiles via carbon–oxygen bond cleavages to form benzophenone dithiolate and hydroquinone. Treatment with organic halides converted only the former products to afford various dithiofunctionalized benzophenones. The depolymerization proceeded as a solid–liquid reaction in the initial phase. Therefore, this method was not affected by the shape of polyetheretherketone, e.g., pellets or films. Moreover, this depolymerization method was applicable to carbon- or glass fiber-enforced polyetheretherketone material. The depolymerized product, dithiofunctionalized benzophenones, could be converted into diiodobenzophenone, which was applicable to the polymerization.

[1] Interdisciplinary Research Center for Catalytic Chemistry (IRC3), National Institute of Advanced Industrial Science and Technology (AIST), Tsukuba Central 5, 1-1-1 Higashi, Tsukuba, Ibaraki 305-8565, Japan. [2] PRESTO, Japan Science and Technology Agency (JST), 1-1-1 Higashi, Tsukuba, Ibaraki 305-8565, Japan. [3] Department of Applied Physics, Graduate School of Engineering, Osaka University, 2-1 Yamadaoka, Suita, Osaka 565-0871, Japan. [4] Research Institute for Sustainable Chemistry, National Institute of Advanced Industrial Science and Technology (AIST), Tsukuba Central 5, 1-1-1 Higashi, Tsukuba, Ibaraki 305-8565, Japan. ✉email: yasu-minami@aist.go.jp

For a long time, considerable effort has been dedicated to the development of technologies for reusing and recycling plastic materials[1,2]. Particularly, thermal and material recycling has been focused on and developed. In recent years, chemical recycling is becoming increasingly important as a way of chemically converting plastic materials into raw organic substrates and organic functional materials[3–12]. Gasification of plastic wastes is the typical protocol. Efforts have been devoted to the research and development of the monomerization of commodity plastics and engineering plastics such as polyethylene terephthalate and polycarbonate[13–15]. Recently, studies on stable engineering plastics such as polyamides and polyurethanes using transition metal catalysts have been actively conducted to provide low-weight molecules[16,17]. Additionally, the development of polymers containing degradable parts or functionalities to be converted into low-weight molecules is being actively pursued[18–25].

Thus, many studies on the chemical recycling of different plastics are ongoing. On the other hand, super engineering plastics such as polyetheretherketone (PEEK), polyphenylene sulfide (PPS), polysulfone (PSU), and polyethersulfone (PESU) known as commercially available high-performance thermoplastic resins exhibit high thermal stability and mechanical strength (Fig. 1a). Particularly, PEEK is a semicrystalline thermoplastic with outstanding characteristics such as chemical resistance, high melting point, and insolubility in organic solvents, in addition to the aforementioned thermal stability and high mechanical strength[26]. However, these advantages hinder its depolymerization to produce monomers and related low-weight molecules. Currently, the existing depolymerization studies involve only PPS and PESU (Fig. 1b)[27–32], indicating that there are no generic depolymerization methods for these resins. This will constitute a significant environmental burden in the future. Additionally, discarding such high-priced products would result in a significant economic loss. Thus, versatile depolymerization methods for super engineering plastics are in high demand. To overcome this challenge, the development of reaction methodologies to approach insoluble chemicals and cleave stable carbon–oxygen bonds is required. This is because the existing reaction formats with small soluble molecules are not directly applicable. Actually, based on reported conditions[33,34], amination reactions via carbon–oxygen bond cleavage were inapplicable to PEEK (see Supplementary Fig. 1).

The ideal depolymerization is to regenerate the original monomer, i.e., halogenated monomers. However, this is thermodynamically impossible. Thus, we focused on sulfur nucleophiles. Sulfur can be introduced into various organic compounds, and the sulfur functional group can be converted into sulfonium salts as useful eliminating groups[35,36]. Using a transition metal catalyst, both the sulfonium group and the original sulfur functional group can be used as leaving groups to form new bonds[37]. Therefore, if super engineering plastics can be depolymerized and functionalized using sulfur reactants, the prevailing problem can be solved. It is known that a thiolate anion, an active form of the sulfur reactant, can interact with electron-deficient arenes to form an electron-donor–acceptor (EDA) complex[38–40]. We expected that this phenomenon would be effective for the interaction of the main chain of the insoluble PEEK molecular surface, and promote the cleavage of the main chain.

Herein, we report the depolymerization of insoluble PEEK (the robust super engineering plastic) using sulfur nucleophiles to afford two monomers without collapsing the molecular architecture of the main chain. This depolymerization comprised a carbon–oxygen main-chain cleavage and an aryl thiolate

**Fig. 1 Depolymerization of super engineering plastics. a** Examples of super engineering plastics. **b** Reported depolymerization of PPS or PESU to produce low-weight molecules. **c** This work: PEEK depolymerization using sulfur nucleophiles to afford two anionic monomer intermediates, followed by selective functionalization with organic halides produced dithiofunctionalized benzophenones and hydroquinone. NMP, N-methyl-2-pyrrolidone; SingaCycle-A1, chloro[[1,3-bis(2,6-diisopropylphenyl)imidazol-2-ylidene](N,N-dimethylbenzylamine)palladium(II)]; IcHex·HCl, 1,3-dicyclohexylimidazolium chloride; cod, 1,5-cyclooctadiene; dcype, 1,2-bis(dicyclohexylphosphino)ethane; R–X, organic halide.

generation sequence. The sulfur nucleophiles became a source of sulfur anions toward a benzophenone monomer block (Fig. 1c). The subsequent introduction of carbonaceous groups on sulfur with organic halides yielded dithiofunctionalized benzophenone, with the remaining hydroquinone monomer intact. Using 2-phenylethanethiol as the sulfur nucleophile, we successfully achieved a high yield of two monomers. The utility of this unique methodology is that it was not affected by the forms of the PEEK and additives such as glass fibers.

## Results and discussion
**Optimization of the reaction conditions**. The bond dissociation energy of carbon–oxygen bonds is higher than that of carbon–sulfur bonds[41]. In fact, nucleophilic etherification of methylthio-substituted benzonitriles was reported[42]. Therefore, the substitution reaction from oxygen to sulfur on carbon appeared difficult. However, we evaluated the efficacy of the exchange reaction from oxygen to sulfur functional groups applying model substrates, 4-(4-phenoxyphenyl)benzophenone and sodium sulfide, using density functional theory (DFT) calculations (See supplementary Fig. 15). The calculated relative free Gibbs energy was −91.7 kcal/mol between these substrates and the products, 4-sodium benzophenone 4-thiolate and sodium 4-phenoxyphenolate as a thermodynamically much stable product, showing that this process was thermodynamically favorable.

Thus, we applied 2 equiv. of sodium sulfide for the depolymerization of PEEK as a powder ($M_w$ ~20,800 and $M_m$ ~10,300) with 1,3-dimethyl-2-imidazolidinone (DMI) as a solvent at 150 °C for 17 h, followed by quenching with iodomethane. In fact that PEEK is insoluble in most organic solvents even at high temperatures, we confirmed that PEEK did not dissolve and swell in DMI at 150 °C. However, depolymerization proceeded to afford the desired monomer product, 4,4′-dimethylthiobenzophenone (**5**), in a small yield (10%), accompanied by comonomer **4** (44% yield) (Fig. 2a). This result showed that the carbon–oxygen bond on the benzophenone unit was selectively cleaved, and the intermediate comonomer, 4-[4-{4-NaSC$_6$H$_4$C(O)}C$_6$H$_4$O]C$_6$H$_4$ONa (**2**), and monomer, (4-NaSC$_6$H$_4$)$_2$CO (**3**) were formed in situ. Following methylation occurred at sodium thiolates in preference to sodium arylates. The aryloxylate group (—OC$_6$H$_4$ONa) in **2** was not a suitable eliminating group; therefore, the formation of the dithiolate **3** was suppressed (see Supplementary Table 1).

In considering approaches to increase the yield of **5**, we focused on a more reactive aliphatic thiolate nucleophile for the formation of the dithiolate **3**. This thiolate nucleophile has a dual role, exchanging the aryloxy group on the benzophenone unit for an alkylthio group and eliminating the alkyl group from the introduced alkylthio group[43,44] to furnish a sodium thiolate (Fig. 2c). As a result, the use of 2-phenylethanethiol (4 equiv.) with sodium *tert*-butoxide (NaO*t*-Bu) (4 equiv.) was effective, affording **5** in 88% yield (Fig. 2b, Entry 1). This depolymerization proceeded smoothly in the air (see Supplementary Table 4, Entry 2). On the other hand, when *n*-hexanethiol was used, the depolymerization itself proceeded smoothly, but gave **5** in a lower yield (51%) and an intermediate, 4-hexylthio-4′-methylthio-benzophenone (**6**) in 38% yield (Fig. 2b, Entry 2). In contrast, benzylthiol did not undergo complete depolymerization, leaving **4** at a moderate yield (Fig. 2b, Entry 3). Other conditions using *n*-hexanethiol or benzylthols were screened but **5** was not obtained in higher yield with 2-phenylethanethiol (see Supplementary Tables 2 and 3).

Thus, 2-phenylethanethiol exhibited good performance for this depolymerization of PEEK, but **4** remained at a low yield under the conditions. We assumed that the in situ-generated —OC$_6$H$_4$ONa group can activate thiols and be transformed into an aryl alcohol group (—OC$_6$H$_4$OH) as a better-eliminating group (Fig. 2d and

Supplementary Fig. 16). In fact, the arylate group (e.g., pKa of PhOH in DMSO: 18.0)[45] exhibits sufficient basicity to activate the aliphatic thiols (e.g., pKa of *n*-BuSH in DMSO: 17.0)[46]. This was supported using DFT calculations (see Supplementary Fig. 17). Successfully, the depolymerization with 3 equiv. of NaO*t*-Bu led to the production of **5** in high yield (84% isolated yield) (Fig. 2b, Entry 4), whereas changing to 2 equiv. of the base diminished the yield of **5**, probably due to decreased reaction rate (Fig. 2b, Entry 5). Using NaOH as a commodity and cheap base instead of NaO*t*-Bu, the yield of **5** was maintained (Fig. 2b, Entry 6). Finally, this depolymerization of PEEK in *N,N*-dimethylacetamide afforded **5** in 93% isolated yield while regenerating hydroquinone (**7**) in more than 95% NMR yield (Fig. 2b, Entry 7). At this time, di(2-phenylethyl)sulfide and styrene was formed in 94% and 4% yields based on the amount of the thiol, respectively, which can be reconverted into the original thiol[47,48]. Notably, at the ambient temperature (30 °C), no monomer and comonomer products were observed (see Supplementary Table 4, Entry 15). Other thiols, such as 2-mercaptoethanol and 2-ethylhexyl-3-mercaptopropanoate containing easily removable carbonaceous groups[49,50] reduced the yields of **5** (see Supplementary Table 3, Entries 16–19).

To examine the efficiency of this depolymerization, we monitored the reaction of PEEK powder with 2-phenylethanethiol under certain conditions shown in Fig. 2b, Entry 4. Unexpectedly, the depolymerization of the insoluble PEEK was practically completed after 1 h, affording **5** and 4-phenethylthio-4′-methylthio-benzophenone (**8**) in 25% and 60% yields, respectively, with small amounts of comonomers (Fig. 2e). Intermediate **8** was gradually converted into **5**. This showed that 2-phenylethyl thiolate is effective for the facile depolymerization of PEEK solids, giving the monomer intermediates speedily. In case of using Na$_2$S, the cleavage of the main chain of PEEK itself was indicated to proceed rapidity (see Supplementary Fig. 8 and 9). With this observation, we expected that this method is insulated from the influence of the PEEK form. In fact, both pellet (average Mw ~20,800; average Mn ~10,300; mean particle size, 80 microns) and film (thickness, 0.025 mm) forms of PEEK smoothly underwent depolymerization sequence to afford **5** and **7** in excellent yields (Fig. 2f). It is noteworthy that the PEEK pellets did not dissolve and divide into small parts but were continuously dwindling during this depolymerization (see Supplementary Table 5 and Fig. 7). This showed that the surface moieties of the PEEK materials reacted with the thiolate without dissolving.

**Substrate scope**. To demonstrate the scope of this one-pot protocol, we examined various electrophiles after the depolymerization of PEEK powder under optimal conditions (Fig. 2b, Entry 7). As shown in Fig. 3, various alkyl halides were applicable in this sequence, and corresponding benzophenone-derived monomer products **9-22** were isolated at excellent yields. For example, 2-bromoethanol underwent the alkylation to form **17** with two hydroxy groups at a good yield[51,52], showing that the hydroxy group did not interfere with this process. 2-(Bromomethyl)oxirane was uneventfully applicable to this process to form **18**, which is a sulfur analog of a benzophenone-based monomer of epoxy resins[53–56]. Notably, the depolymerization/functionalization sequence could be easily performed on a gram scale. The use of 1-bromo-3,7-dimethyloctane was selected in the gram scale sequence, which afforded **19** in 81% yield. In addition to alkyl halides, an acid chloride was applicable in this sequence to form a corresponding monomer product, **20** in 81% yield. Further treatment with hydrogen chloride after depolymerization afforded 4,4′-dimercaptobenzophenone (**21**)[57]. This one-pot depolymerization/functionalization method was applicable in the three-step sequence. After treatment with 2-bromoethanol, in situ formed **17** was subjected to esterification with methacryloyl

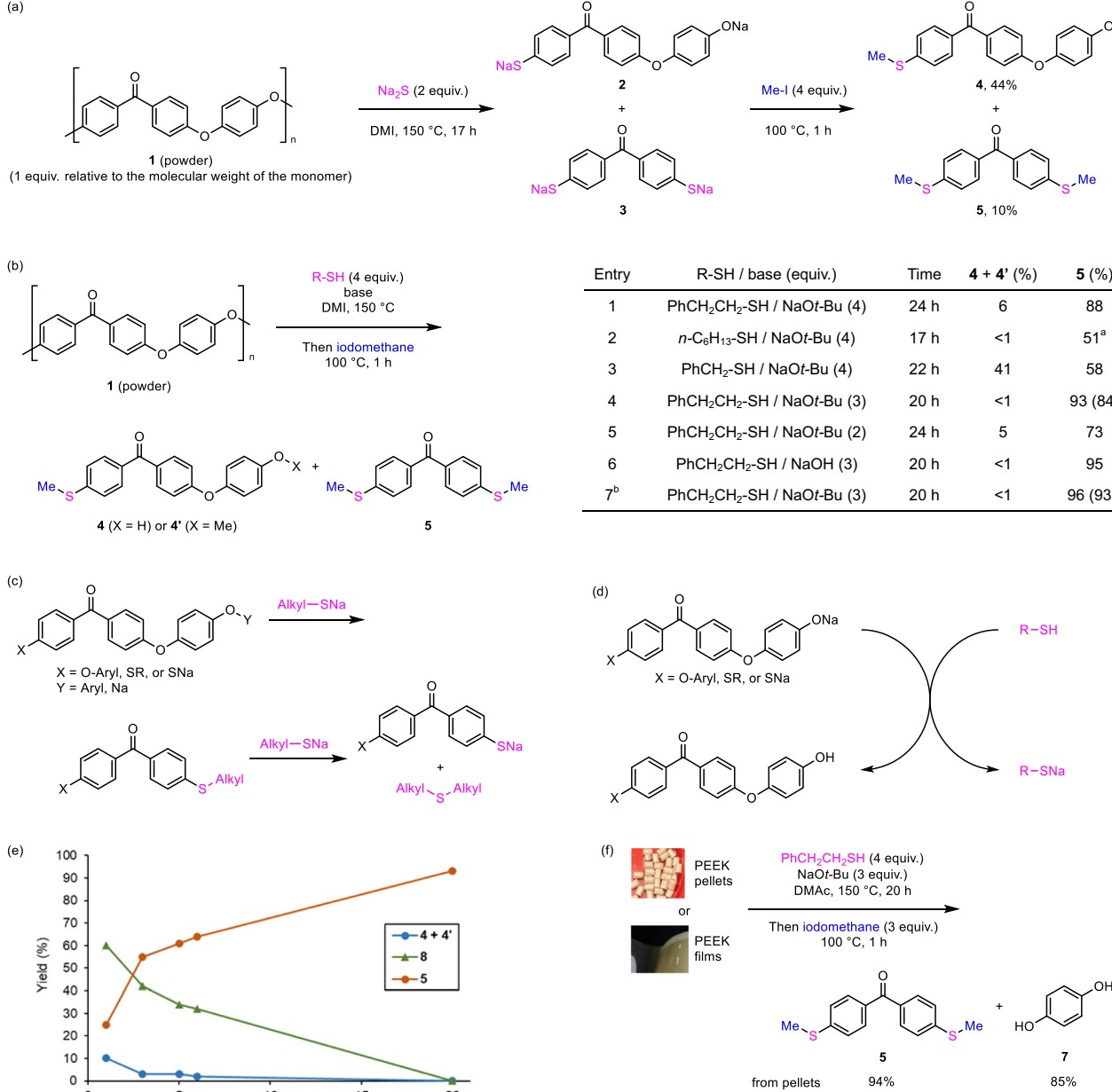

**Fig. 2 Development of PEEK depolymerization methodology. a** Reaction of PEEK with sodium sulfide followed by the treatment with iodomethane. **b** Development of PEEK depolymerization. Yields were determined via gas chromatography and nuclear magnetic resonance (NMR). The numbers in parentheses are isolated yields. **c** Cleavage of carbon–oxygen bond on the benzophenone unit by sodium alkylthiolates to form *para*-alkylthio-benzophenone units, followed by the nucleophilic attack of the sodium alkylthiolates to the alkyl group on sulfur. **d** Exchange between intermediate sodium arylates and the thiols to provide aryl alcohols and sodium thiolates. **e** Time-dependent conversion for the reaction of PEEK powder (1 equiv. relative to the molecular weight of the monomer), PhCH₂CH₂SH (4 equiv.), NaOt-Bu (3 equiv.), and DMI at 150 °C, followed by quenching with iodomethane at 100 °C. **f** Depolymerization of PEEK pellets or films. [a]4-Hexylthio-4′-methylthio-benzophenone was observed in 38% yield. [b]N,N-dimethylacetamide was used as a solvent. [c]Hydroquinone (**7**) was obtained in high yield (>95%) determined by ¹H NMR analysis. DMAc N,N-dimethylacetamide.

chloride, and the corresponding product **22** reported as a monomer for high refractive index resin was easily obtained[51,52].

**Selective depolymerization of PEEK.** Given that the resin frequently contains an additional agent, we investigated the depolymerization of PEEK powder in the presence of glass fibers. As shown in Table 1, Entry 1, the depolymerization proceeded smoothly to form **5** and hydroquinone (**7**) in 94 and 92% yields, respectively. Additionally, other commodity polymers such as polypropylene, polystyrene, and Nylon 6 did not interfere with

the depolymerization, affording **5** and **7** in good yields (Table 1, Entries 2−4). Thereafter, carbon or glass fiber-reinforced PEEK material was used for the depolymerization experiment. A roughly ground carbon fiber (30 wt%) reinforced PEEK material was subjected to a reaction with 2-phenylethanethiol and NaOt-Bu, followed by treatment with iodomethane, affording **5** and **7** in good yields (Table 1, Entry 5). Similarly, a roughly ground PEEK compound including glass fiber (30 wt%) was examined under the same conditions, also resulting in **5** and **7** (Table 1, Entry 6).

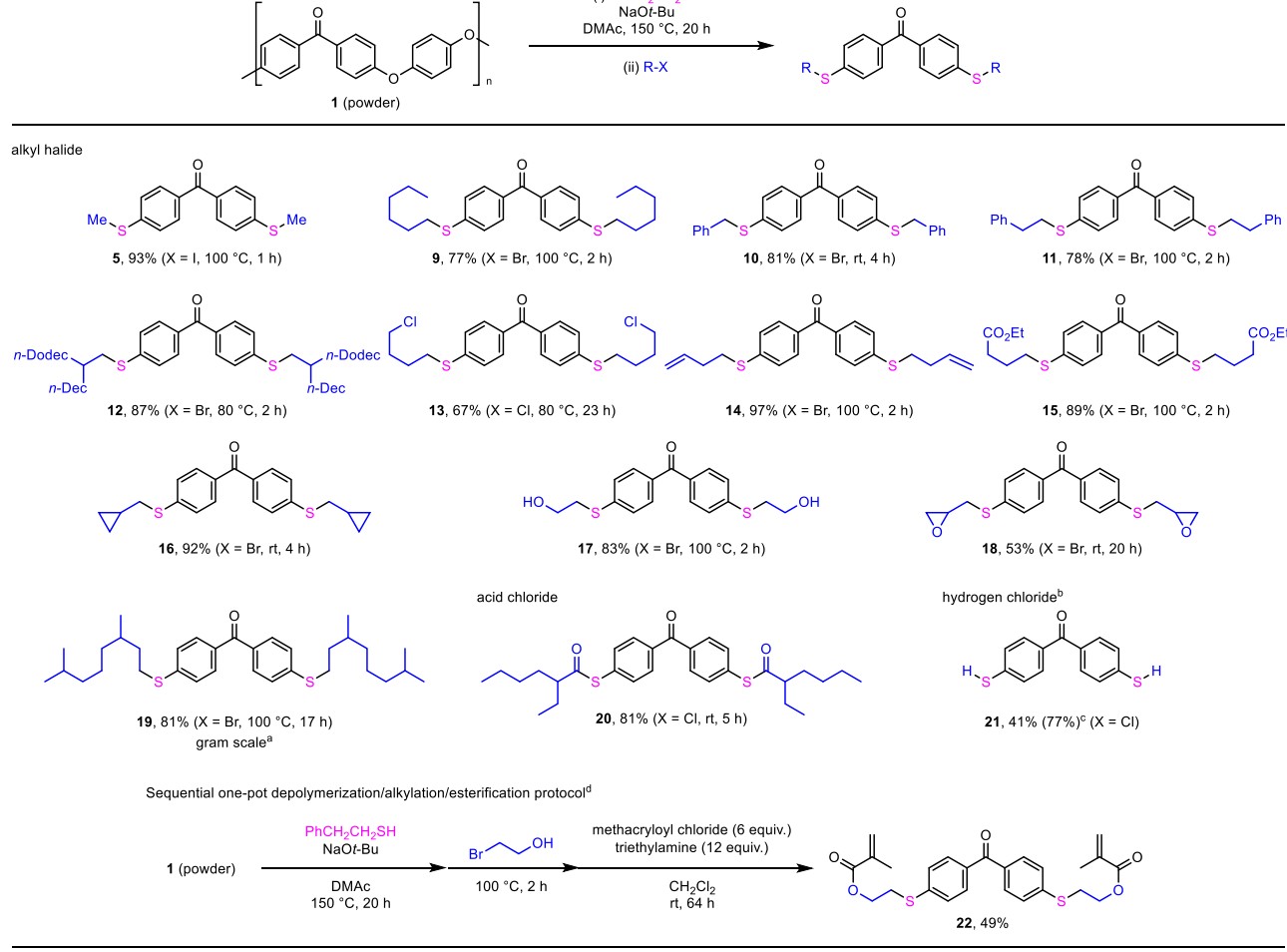

**Fig. 3 Scope of electrophiles after PEEK depolymerization.** Reaction conditions: (i) PEEK powder (0.3 mmol relative to the molecular weight of the monomer), PhCH$_2$CH$_2$SH (1.2 mmol), NaO$t$-Bu (0.9 mmol), DMAc (0.6 mL), 150 °C, 20–22 h. (ii) Electrophile (0.9 mmol) under conditions in parentheses. For each compound, the isolated yield is given in percentage. [a]PEEK powder (4.0 mmol relative to the molecular weight of the monomer), PhCH$_2$CH$_2$SH (16 mmol), NaO$t$-Bu (12 mmol), DMAc (8 mL), 150 °C, 22 h. (ii) 1-bromo-3,7-dimethyloctane (12 mmol), 100 °C, 2 h. [b]aq. HCl (2 M, 2.0 mL). [c]NMR yield. [d]After step (ii), methacryloyl chloride (1.8 mmol), triethylamine (3.6 mmol), and dichloromethane (0.6 mL) was added to the mixture, which was stirred at room temperature for 64 h.

**Experimental mechanistic studies.** As previously mentioned, 2-phenylethanethiol is a promising depolymerization reagent for PEEK. To understand this efficiency, we examined the reaction of 4,4′-diphenoxy-benzophenone (**23**) as a PEEK model compound with 2 equiv. of selected thiols and NaO$t$-Bu at 150 °C for 1 h. First, the reaction using $n$-hexanethiol formed di($n$-hexylthio)-substituted benzophenone (**9**) as a simple substitution product and following dehexylated/monomethylated **6** in 70% and 22% yields, respectively (Table 2, Entry 1). Benzyl mercaptan afforded dimethylated product **5** in 17% yield with the observation of dibenzyl sulfide (32% based on the amount of the thiol) (Table 2, Entry 2). These results were due to the stability toward nucleophilic attack; hexyl group is robust, whereas the benzyl group is weak. In contrast, the reaction using 2-phenylethanethiol furnished a mixture of monomethylated **8**, **5**, and styrene in 67, 26, and 37% yields, respectively, without the generation of bis(2-phenylethyl)sulfide (Table 2, Entry 3). For longer reaction time, the yield of **5** was increased to 70% (see Supplementary Fig. 2). At room temperature, this reaction took place to afford **11** selectively in 95% yield (see Supplementary Fig. 3). The generation of styrene means that base-mediated elimination from the phenethyl group on **8** and **11** occurs gradually to provide sodium thiolates[58]. Thus, these results show that 2-phenylethanethiol can lead to the

formation of (4-NaSC$_6$H$_4$)$_2$CO (**3**) effectively via the smooth carbon–oxygen bond-cleaving substitution followed by two types of dealkylation; second nucleophilic substitution (Fig. 2c) and styrene elimination. In fact, we confirmed that these dealkylations proceed (see Supplementary Fig. 6).

As mentioned above, the present depolymerization was initiated on the surface of the PEEK materials using the thiolate. This suggestion was supported by the S $K$-edge X-ray absorption near-edge structures (XANES) analysis of degradation samples prepared by the reaction of PEEK powder with sodium sulfide at the early stage (see Supplementary Figs. 10, 11). A thiolate anion and an electron-deficient arene such as benzophenone are known to associate to form an EDA complex (see Supplementary Figs. 12–14)[59]. We expected that this EDA complex would retain the thiolate anion on the PEEK surface and promote the surface carbon–oxygen bond cleavage, maybe via the S$_N$Ar or S$_{RA}$1[60–62] mechanism. In the case of S$_{RA}$1, radical-chain and nonchain mechanisms are proposed. In this regard, this PEEK depolymerization provided hydroquinone, which is known as an inhibitor of the generation of free radicals, suggesting that free radicals were not generated during the depolymerization (see Supplementary Fig. 4, 5). In addition, 4 equiv. of (2,2,6,6-tetramethylpiperidin-1-yl)oxyl (TEMPO), as a radical scavenger, did not affect the depolymerization for 3 h and methylation, giving **5** and

**Table 1 Depolymerization of PEEK in the presence of additives or reinforced PEEK.**

| Entry | PEEK | Additive | Time (h) | 5 (%) | 7 (%) |
|---|---|---|---|---|---|
| 1 | Powder | glass fiber | 20 | 94 | 92 |
| 2 | Pellets | polypropylene | 40 | 85 | 72 |
| 3 | Pellets | polystyrene | 40 | 85 | 70 |
| 4 | Pellets | Nylon 6 | 40 | 83 | 69 |
| 5 | 30 wt% Carbon fiber-reinforced PEEK (Roughly ground) | - | 20 | (65)[a] | 71 |
| 6 | 30 wt% Glass fiber-reinforced PEEK (Roughly ground) | - | 20 | (53)[b] | 73 |

Reaction conditions: (i) PEEK (0.3 mmol relative to the molecular weight of the monomer), 2-phenylethanethiol (1.2 mmol), NaOt-Bu (0.9 mmol), DMAc (0.6 mL), 150 °C, 1 h. For each compound, the NMR yield is given in percentage. The numbers in parentheses are isolated yields.
[a] 8 was obtained in 19% isolated yield.
[b] 8 was obtained in 20% isolated yield.

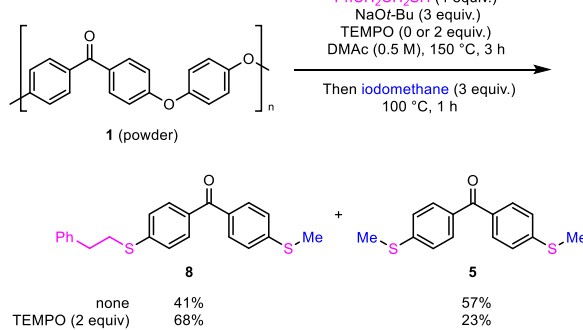

Fig. 4 Depolymerization of PEEK at the early stage. TEMPO does not affect depolymerization.

8 in good yields whereas the yield of 5 was lower than the TEMPO-free conditions (Fig. 4). These results might at least allow us to exclude the possibility of a radical pathway under the depolymerization step by alkylthiolates.

**Proposed mechanism.** Based on the experimental results, we show a proposed depolymerization pathway in Fig. 5. The benzophenone unit at the PEEK surface and an organic thiolate derived from the corresponding thiol with NaOt-Bu first associate to form an EDA complex. The sulfur center of the thiolate then attacks the *para*-carbon bound to oxygen via the $S_NAr$ or nonchain $S_{RN}1$ mechanism, and the aryloxy anion is released to complete the formation of the carbon–sulfur bond. The generated aryloxy anion or NaOt-Bu activates the thiol to form the organic thiolate, which undergoes an $S_N2$ reaction with the generated alkyl aryl sulfide to produce the benzophenone thiolate moiety and a dialkylsulfide. In the case using 2-phenylethanethiol, the base-mediated elimination of styrene from 2-phenylethylthio group also proceeds sluggishly to form the thiolates and styrene. This series of processes repeatedly occurred to finally obtain the benzophenone dithiolate 3.

**Utility of products.** Alkylthio groups can be converted into reactive sulfonium groups. We confirmed the methylation of 5 using methyl trifluoromethane sulfonate in 1,2-dichloroethane at 60 °C, based on a reported method[63], and obtained the benzophenone 4,4′-bis(di-methylsulfonium) salt, 24 in excellent yield (Fig. 6a). Afterward, we attempted substitution to iodine. Based on a reported nickel catalytic method developed by Yorimitsu[64], 24 was converted into 4,4′-diiodobenzophenone (25) as an active form of various substitution reactions[65–67]. In fact, 25 was applicable to the polymerization with 2,2′-bis(4-hydroxyphenyl)propane (26) under copper-catalyzed conditions[68] to give the corresponding copolymer 27[69–72] with $M_w = 24,039$ and PDI = 3.49 in 87% yield after reprecipitation (Fig. 6b). We also examined the polymerization using molecules obtained by the present depolymerization (see Supplementary pages S22–S24). The reaction of 4,4′-dimercaptobenzophenone (21) with nonanedioyl dichloride (28) underwent in chloroform under reflux[73] to form a polythioester 29 with $M_w = 49,641$ and PDI = 2.04 in 94% yield (Fig. 6c).

## Conclusion

In this study, we demonstrated that insoluble PEEK, as a robust super engineering plastic, can be depolymerized for the formation of monomer units. In this process, the 2-phenylethanethiolate reagent was effective for the depolymerization of PEEK, followed by treatment with organic halides to furnish dithiofunctionalized benzophenones and hydroquinone in high yields. A series of organic halides were applied after the depolymerization, providing various dithiofunctionalized benzophenones. The products can be

**Table 2 Examination of the effect of thiols using a model substrate.**

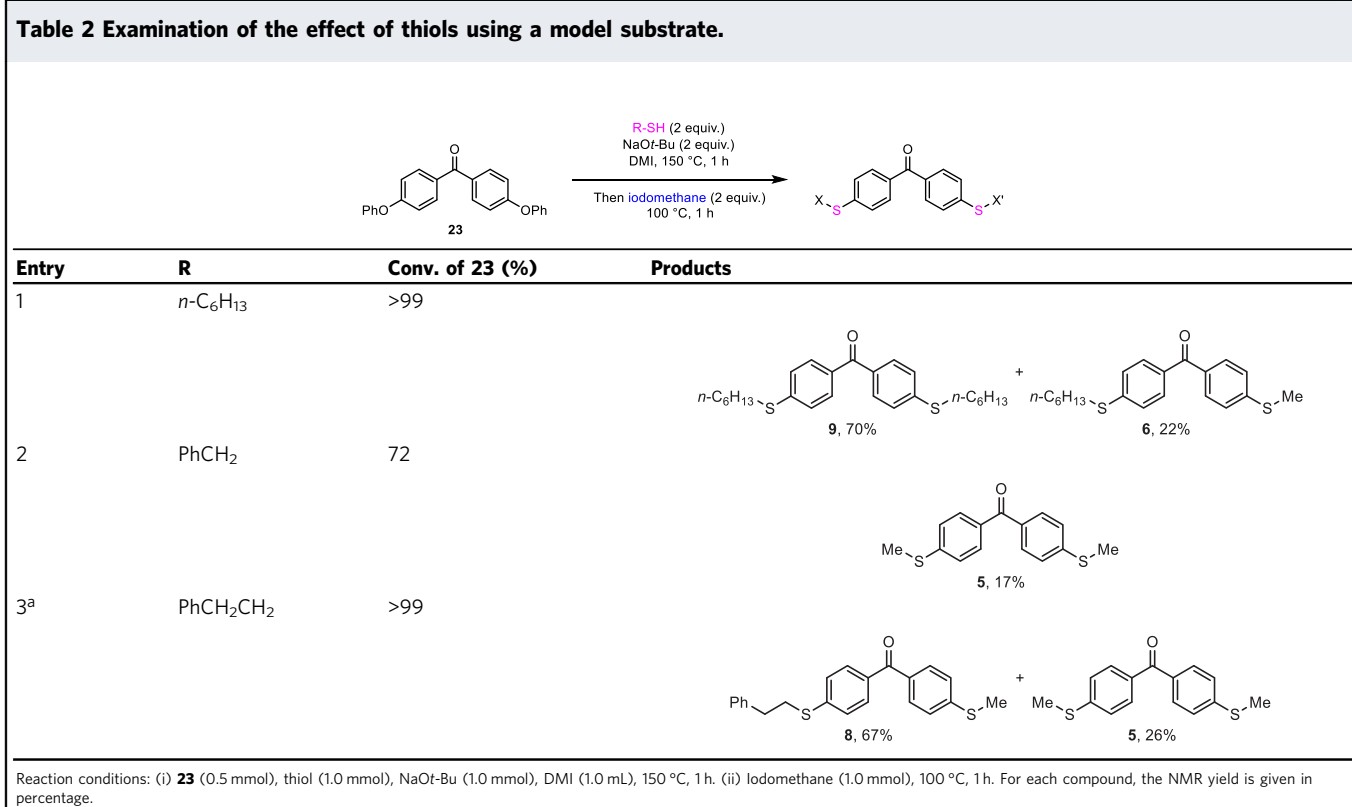

| Entry | R | Conv. of 23 (%) | Products |
|---|---|---|---|
| 1 | $n$-C$_6$H$_{13}$ | >99 | **9**, 70% + **6**, 22% |
| 2 | PhCH$_2$ | 72 | **5**, 17% |
| 3[a] | PhCH$_2$CH$_2$ | >99 | **8**, 67% + **5**, 26% |

Reaction conditions: (i) **23** (0.5 mmol), thiol (1.0 mmol), NaO$t$-Bu (1.0 mmol), DMI (1.0 mL), 150 °C, 1 h. (ii) Iodomethane (1.0 mmol), 100 °C, 1 h. For each compound, the NMR yield is given in percentage.
[a]Styrene was generated in 37% yield based on the amount of the thiol.

**Fig. 5 Proposed depolymerization of PEEK.** Cleavage of carbon–oxygen main chains by the alkyl thiolate followed by dealkylation on sulfur to form aryl thiolate generation. Sodium cation is omitted.

converted into a bis(sulfonium) salt and diiodobenzophenone as an active form of various substitution reactions. Several produced monomer units were applied to polymerization reactions. The depolymerization proceeded as a solid–liquid reaction in the initial phase. Therefore, the present depolymerization method was applicable to various forms of pure PEEK, such as powder, pellet, and film. Moreover, glass or carbon fiber-reinforced PEEK materials were utilized for this depolymerization. This development opens up the application of PEEK in chemical recycling and

highlights the potential of this strategy to unlock the depolymerization of other highly stable resins. Further efforts are underway to exploit the catalytic methodology for the depolymerization of PEEK and to expand the scope of other super engineering plastics and related robust polymer materials.

## Methods

**General procedure for depolymerization of PEEK.** *N,N*-Dimethylacetamide (0.60 mL) and 2-phenylethanethiol (167 mg, 1.21 mmol) were added to a mixture

**Fig. 6 Utility of products. a** Transformation of 4,4′-di(methylthio)benzophenone (**5**) to benzophenone 4,4′-bis(dimethylsulfonium) salt **24** followed by iodination to form 4,4′-diiodobenzophenone (**25**). **b** Polymerization of **25** with bis-phenol A (**26**). **c** Polymerization of **21** with nonanedioyl dichloride (**28**). MeOTf methyl trifluoromethane sulfonate, glyme dimethoxyethane; Neocuproine, 2,9-dimethyl-1,10-phenanthroline, DMF N,N-dimethylformamide.

of PEEK powder (86.4 mg, 0.300 mmol relative to the molecular weight of the monomer) and sodium *tert*-butoxide (86.5 mg, 0.900 mmol) in a 3 mL vial in an argon atmosphere. The mixture was stirred at 150 °C for 20 h. After the liquid mixture cooled to room temperature, iodomethane (128 mg, 0.900 mmol) was added and stirred at 100 °C for 1 h. After ethyl acetate (1.5 mL) was added, the mixture was washed with aqueous HCl (2 M, 1 mL), water, and brine. At this time, the mixture was analyzed by $^1$H NMR to determine the yields of hydroquinone (**7**) (>95%) and styrene (4%). The extracted organic layer was dried over Mg$_2$SO$_4$ and concentrated in vacuo. The crude product was purified by column chromatography on silica gel (hexane/ethyl acetate, 96:4 to 7:3) to afford bis(4-(methylthio)phenyl) methanone (**5**) (93%, 75.9 mg).

**General information**. See Supplementary Methods, general information (page S3).

**Chemicals**. See Supplementary Methods, chemicals (page S4).

**NMR charts**. See Supplementary Data 1, NMR spectra of obtained chemicals.

**GPC charts**. See Supplementary Data 1, GPC charts.

## Data availability

The data obtained in this study are available within this article and its supplementary information and are also from the corresponding authors upon reasonable request. Original $^1$H and $^{13}$C spectra of the compounds obtained in this manuscript are available in Supplementary Data 1. The computed energy values and coordinates are available in Supplementary Data. 2.

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

## Acknowledgements

This work was supported financially by PRESTO (JPMJPR21N9 to Y.M.) from the JST, Fujimori Science and Technology Foundation, Iketani Science and Technology Foundation, Grants-in-Aid for Scientific Research (C) (19K05481 to Y.M.) from the JSPS, and Department of Materials and Chemistry, AIST. Y.M., N.M., and Y.N. also acknowledge the DIC Corporation. Y.M. thanks JST, ERATO (JPMJER2103), and Prof. Kyoko Nozaki and her lab members for discussions on this project. Y.M. thanks Dr. Shinji Tanaka for the solid-state NMR analysis. A part of this work was performed under the approval of the Photon Factory Program Advisory Committee (Proposal No. 2021PF-G021). We would like to thank Dr. Shinji Tanaka for his contribution to the analysis of the polymer products and Ms. Tomoo Tsuyuki, Ms. Risa Kawato, and Mr. Yuki Inagaki for their kind assistance in experiments. We dedicated ourselves to Professor Shigeru Yamago on the occasion of his 60th birthday.

## Author contributions

Y.M. conceived the idea and designed the whole experiment with N.M. N.M. also worked with Y.M. to plan the conversion of the products and polymer synthesis. Y.M., N.M., and M.S. performed the experiments. Y.T. carried out XANES analysis. R.W. carried out analytical pyrolysis experiments. Y.M., N.M., Y.T., R.W., and Y.N. contributed to writing the manuscript and participated in data analyses and discussions. Y.M., N.M., and Y.N. revised the paper.

## Competing interests

The authors declare no competing interests.
