## [Peer Review File · Communications Chemistry]

Reviewers' comments:

Reviewer #1 (Remarks to the Author):

In this contribution, the depolymerization of insoluble PEEK was reported using sulfur nucleophiles via carbon–oxygen bond cleavages to form benzophenone dithiolate and hydroquinone. Various dithiofunctionalized benzophenones were obtained through this method. This was an interesting concept because it provided an effective method for depolymerization of engineering plastics. I would recommend it for publication if the following questions are resolved.

1. In page 5, the author claimed that the aryloxy group ($-\text{OC}_6\text{H}_4\text{ONa}$) in 2 was not a suitable eliminating group; therefore, the formation of the dithiolate 3 was suppressed. However, in Fig. S14, under the reaction of Na_2S , the free Gibbs energy of $-\text{OC}_6\text{H}_4\text{ONa}$ leaving was -35.9 kJ/mol, which was thermodynamically favorable. Does this mean that under the reaction of excess Na_2S , the selectivity of depolymerization could be improved? When thiolate nucleophile was used as depolymerization reagent, the added amount was 4 equivalents. Would the equivalent of thiol affect the reaction process?
2. In Fig. 2, the yield and formation of hydroquinone were not mentioned at all. I want to know whether the formation of hydroquinone could be observed through gas chromatography and nuclear magnetic resonance in the reaction process, for example, in Fig. 2a.
3. It seemed that the optimal condition was when the amount of base added was 3 equivalents. Could the author give the possible reasons why both 2 and 4 equiv. would inhibit the improvement of product 5 yield?
4. The author pointed out that the reaction was carried out at the solid-liquid interface, and the shape of raw materials did not affect the reaction. From Fig. 2e, we could know that PEEK powder would be completely dissolved in 1 h, so did bulky PEEK dissolve so quickly as well? In addition, Fig. 2e should add a few more points, especially after 5 h.
5. In Table 1, the depolymerization/functionalization process of the one-pot method was realized to generate 22. Is there any special purpose and function of 22?
6. In Fig. 4, the possible mechanism was given. I wonder if the key steps could give more evidence. For example, in Fig. S3, 11 was specifically generated. Whether the interaction between 11 and base or thiol could be explored to further prove the generation of the product 3.
7. In Fig. 5, the yield of iodide 25 was only 25%. Did the author try to further optimize the reaction conditions to obtain a higher yield? In addition, difluorodibenzophenone was usually used as a monomer of PEEK. Could fluoride be obtained by this method?
8. The chemical recycling to monomers (CRM) is of great significance. Therefore, some related references should be included in the introduction part, such as ACS Catal. 2019, 9, 8012–8067; Green Chem. 2022, 24, 2321–2346; Chem. Eur. J. 2018, 24, 11255–11266.

Reviewer #2 (Remarks to the Author):

In this manuscript, the authors demonstrated the depolymerization of poly(ether ether ketone) using sulfur nucleophiles to form benzophenone dithiolate and hydroquinone. The depolymerization product, dithiofunctionalized benzophenones can be converted to reactive sulfonium groups, which can be further functionalized into useful building blocks. Through screening of conditions, the authors identified 2-phenylethane thiol as an optimal nucleophile for depolymerization. The authors proposed a reasonable depolymerization pathway that involves $\text{S}_\text{N}\text{Ar}$ reaction followed by an $\text{S}_\text{N}2$ to generate benzophenone dithiolate and dialkylsulfide. Different electrophiles can be used to react with benzophenone dithiolate, and the process shows excellent functional group tolerance, allowing for the formation of various functional monomers such as diols and bis-epoxides. The depolymerization is

found to be efficient for PEEK of different forms and with additives, making this method applicable to a broad range of materials. This is a comprehensive, high-quality study and should appeal to the readership of Communications Chemistry; I therefore recommend its publication after the authors address the following minor concerns:

1. Since the usage of NaOH as the base also worked comparably well to NaOt-Bu toward a high yield of dithiofunctionalized benzophenone as stated in Fig. 2 entry 6, I am wondering the reasons why the authors did not choose to use NaOH for the following investigation, given that NaOH is more readily available and easier to handle.
2. In the manuscript, the authors referred to over 20 compounds, and their labeling is scattered among the figures, which makes it difficult to track some. It will be helpful if the authors can group all the compounds in one scheme.

Point-by-Point response to referees

All the responses made are summarized in followings.

Comments from Reviewer 1:

1-1) In page 5, the author claimed that the aryloxyate group ($-\text{OC}_6\text{H}_4\text{ONa}$) in **2** was not a suitable eliminating group; therefore, the formation of the dithiolate **3** was suppressed. However, in Fig. S14, under the reaction of Na_2S , the free Gibbs energy of $-\text{OC}_6\text{H}_4\text{ONa}$ leaving was -35.9 kJ/mol, which was thermodynamically favorable. Does this mean that under the reaction of excess Na_2S , the selectivity of depolymerization could be improved? When thiolate nucleophile was used as depolymerization reagent, the added amount was 4 equivalents. Would the equivalent of thiol affect the reaction process?

Response to 1-1)

We appreciate the Reviewer 1 for the valuable comment. We have examined the reaction of PEEK pellet with 4 equiv. of Na_2S (This data was newly added in the Table S1, run 9 in the Supplementary Information). However, the yields and the selectivity of the reaction were not improved. In addition, the reaction shown in Entry 14, Table S1 also employed 4 equiv. of Na_2S , giving a similar result shown in Entry 13, Table S1 using 2 equiv. of Na_2S . Thus, we thought that use of more amounts of Na_2S does not affect the reactivity and selectivity.

1-2) In Fig. 2, the yield and formation of hydroquinone were not mentioned at all. I want to know whether the formation of hydroquinone could be observed through gas chromatography and nuclear magnetic resonance in the reaction process, for example, in Fig. 2a.

Response to 1-2)

We appreciate the Reviewer 1 for the valuable comment. As explained in Fig. 2, the formation of hydroquinone could be detected by ^1H NMR. In fact, hydroquinone was observed in $>95\%$ NMR yield in Entry 7, Fig. 2b. To clarify this information, we revised the last part in the explanation in Fig. 2. Moreover, we have already shown the generation and NMR yields of hydroquinone from the depolymerization of PEEK pellets or films in Fig. 2f as well as the depolymerization of PEEK in the presence of additives or reinforced PEEK in Table 2 in the original manuscript.

1-3) It seemed that the optimal condition was when the amount of base added was 3 equivalents. Could the author give the possible reasons why both 2 and 4 equiv. would inhibit the improvement of product **5** yield?

Response to 1-3)

We appreciate the Reviewer 1 for the valuable comment. In the case using 4 equiv. of NaOt-Bu , the yield of **5** was lower than the case of 3 equiv. as shown in Fig. 2b. At that time, as mentioned in the manuscript, $-\text{OC}_6\text{H}_4\text{ONa}$ group, which is lower elimination activity than $-\text{OC}_6\text{H}_4\text{OH}$, remains during the reaction. For this reason, we think the yield of **5** was lower in the depolymerization using 4 equiv. of the base. On the other hand, we think that the conditions using 2 equiv. of NaOt-Bu reduce the depolymerization rate rather than using 3 equiv. These

explanations were added to the sentence in page 6, line 3-5. We aimed at finding conditions that complete the depolymerization as quickly as possible, so we determined these conditions employing 3 equiv. of NaOt-Bu to be optimal.

1-4) The author pointed out that the reaction was carried out at the solid-liquid interface, and the shape of raw materials did not affect the reaction. From Fig. 2e, we could know that PEEK powder would be completely dissolved in 1 h, so did bulky PEEK dissolve so quickly as well? In addition, Fig. 2e should add a few more points, especially after 5 h.

Response to 1-4)

We appreciate the Reviewer 1 for the valuable comment. We checked the reaction mixture from the depolymerization for 1 h and observed the newly generation of much amount of solid sodium salts including 4-[4-{4-NaSC₆H₄C(O)}C₆H₄O]C₆H₄ONa (**2**), and monomer, (4-NaSC₆H₄)₂CO (**3**). For this reason, it was difficult to find the residual PEEK powder in this reaction mixture. But, considering the yields of generated salts, we assume that the PEEK powder was almost consumed. In this relation, we newly examined the depolymerization of PEEK pellets for 1 h and 6 h and analyzed the yields of **4**, **5**, **7**, and **8**. We also observed a decrease in the size of the remaining PEEK pellets after the reactions. These results were added in the Supplementary Information, page S24, Table S5 and Fig. S7.

We added the result 5 h after the start of the reaction to Fig. 2e and Table S4, Supplementary Information.

1-5) In Table 1, the depolymerization/functionalization process of the one-pot method was realized to generate **22**. Is there any special purpose and function of **22**?

Response to 1-5)

We appreciate the Reviewer 1 for the valuable comment. In chemical recycling, we believe it is effective to eliminate the intermediate isolation process and to obtain the target product smoothly in a continuous process. Therefore, we chose to use a continuous process involving this depolymerization of PEEK, alkylation, and esterification. In addition, compound **17** as the intermediate for the formation of **22** reported as a monomer for high refractive index resin was obtained in high yield by depolymerization followed by alkylation. This result was another reason for adopting the continuous process. In addition, explanation of compound **22** “reported as a monomer for high refractive index resin” was newly added after the phrase “the corresponding product **22**” in page 8.

1-6) In Fig. 4, the possible mechanism was given. I wonder if the key steps could give more evidence. For example, in Fig. S3, **11** was specifically generated. Whether the interaction between **11** and base or thiol could be explored to further prove the generation of the product **3**.

Response to 1-6)

We appreciate the Reviewer 1 for the valuable comment. We examined the reaction of **11** with 2 equiv. of NaOt-Bu in DMI at 150 °C followed by the treatment with iodomethane. As a result, **5** and styrene were formed in 97% and 81% yields, respectively. Moreover, we examined the reaction of **11** with 2 equiv. of both 2-phenylethylthiol and NaOt-Bu under same conditions

and observed the formations of **5**, **8**, styrene, and di(2-phenylethyl)sulfide in 76%, 24%, 34%, and 64% yields, respectively. These results demonstrated the formation of benzophenone dithiolate **3** from **11** by the reaction with base, or thiol and base. These two examinations were added in page 9, line 18 in the manuscript and Page S23, Fig. S6, Supplementary Information.

1-7) In Fig. 5, the yield of iodide **25** was only 25%. Did the author try to further optimize the reaction conditions to obtain a higher yield? In addition, difluorodibenzophenone was usually used as a monomer of PEEK. Could fluoride be obtained by this method?

Response to 1-7)

We appreciate the Reviewer 1 for the valuable comment. In fact, we attempted the substitution from dimethylsulfonium group in **24** to fluorine under the reported conditions employing CsF or KF and Kryptofix® 222. But, the desired fluorinated product, 4,4'-difluorobenzophenone was hardly observed whereas the elimination of methyl group on sulfur proceeded. We thought that the dimethylsulfonium group is not suitable to the direct halogenation reaction. So, we examined the catalytic iodination as shown in Fig. 5. Although we examined various conditions to enhance the yield of **25**, 25% was the highest yield of **25** as shown in Fig. 5.

1-8) The chemical recycling to monomers (CRM) is of great significance. Therefore, some related references should be included in the introduction part, such as ACS Catal. 2019, 9, 8012–8067; Green Chem. 2022, 24, 2321–2346; Chem. Eur. J. 2018, 24, 11255–11266.

Response to 1-8)

We appreciate the Reviewer 1 for the valuable comment. We newly added the related references in the introduction part (above three journal, ACS Catal. 2019, 9, 8012–8067; Green Chem. 2022, 24, 2321–2346; Chem. Eur. J. 2018, 24, 11255–11266 as well as Polym. Chem. 2020, 1, 4830-4849; ChemSusChem 2021, 14, 4041-4070; ChemSusChem 2021, 14, 4123-4136; Beilstein J. Org. Chem. 2021, 17, 589-621).

Comments from Reviewer 2:

2-1) Since the usage of NaOH as the base also worked comparably well to NaOt-Bu toward a high yield of dithiofunctionalized benzophenone as stated in Fig. 2 entry 6, I am wondering the reasons why the authors did not choose to use NaOH for the following investigation, given that NaOH is more readily available and easier to handle.

Response to 2-1)

We appreciate the Reviewer 2 for the valuable comment. As mentioned by the Reviewer 2, NaOH is effective base for this depolymerization. Exactly, NaOH is practically suitable to this depolymerization. But the method using NaOH generates stoichiometric amounts of H₂O, which may react with reactive substrates such as esters, alkyl halides, and epoxides. In other word, the use of NaOH would reduce the versatility of organohalides on the second alkylation step shown in Table 1. In addition, the yield of **5** was slightly lower than the reaction using NaOt-Bu (Fig. 2b, Entry 6 and 7). Thus, we determined the conditions employing NaOt-Bu to be optimal.

2-2) In the manuscript, the authors referred to over 20 compounds, and their labeling is scattered among the figures, which makes it difficult to track some. It will be helpful if the authors can group all the compounds in one scheme.

Response to 2-2)

We appreciate the Reviewer 2 for the valuable comment. We added the compound numbers “**9-22**” after the phrase “corresponding benzophenone-derived monomer products” in page 8. In addition, we depicted the structure of **5** in Fig. 5 and PEEK (**1**) in Table 2 to understand the shown compounds easily. Moreover, to make the comparison of each product easier, the product **5** with the yield and conditions were added to Table 1.

REVIEWERS' COMMENTS:

Reviewer #1 (Remarks to the Author):

The authors address all of the reviewers' questions and the paper can be now accepted for publication.

Reviewer #2 (Remarks to the Author):

The authors have sufficiently addressed my comments. I recommend publication of the manuscript without further change.

Point-by-Point response to referees

All the responses made are summarized in followings.

Comments from Reviewer 1:

The authors address all of the reviewers' questions and the paper can be now accepted for publication.

Response to Reviewer 1

We appreciate the Reviewer 1 for the valuable comment. We would like to thank the reviewer 1 for your consideration.

Comments from Reviewer 2:

The authors have sufficiently addressed my comments. I recommend publication of the manuscript without further change.

Response to Reviewer 2

We appreciate the Reviewer 2 for the valuable comment. We would like to thank the reviewer 1 for your consideration.